# Resistance Mechanisms to Combined CDK4/6 Inhibitors and Endocrine Therapy in ER+/HER2− Advanced Breast Cancer: Biomarkers and Potential Novel Treatment Strategies

**DOI:** 10.3390/cancers13215397

**Published:** 2021-10-27

**Authors:** Abeer J. Al-Qasem, Carla L. Alves, Henrik J. Ditzel

**Affiliations:** 1Department of Cancer and Inflammation Research, Institute of Molecular Medicine, University of Southern Denmark, DK-5000 Odense, Denmark; aalqasem@health.sdu.dk (A.J.A.-Q.); calves@health.sdu.dk (C.L.A.); 2Department of Oncology, Odense University Hospital, Institute of Clinical Research, University of Southern Denmark, DK-5000 Odense, Denmark; 3Academy of Geriatric Cancer Research (AgeCare), Odense University Hospital, DK-5000 Odense, Denmark

**Keywords:** breast cancer, CDK4/6 inhibitor, biomarkers, resistance mechanisms, liquid biopsy

## Abstract

**Simple Summary:**

Advanced stage breast cancer is incurable, and a leading cause of cancer-related death in women. Recently, novel treatment options have improved clinical outcomes of breast cancer patients. Combined CDK4/6 inhibitor and endocrine therapy have been shown to be highly effective, improving both progression-free and overall survival, but resistance inevitably develops and disease progression occurs. Identification of clinically useful biomarkers predicting therapeutic response/failure and resistance mechanisms are essential for optimal patient stratification and the development of novel therapeutic strategies for patients who have progressed on combined CDK4/6 inhibitor and endocrine therapy. However, finding a universal single predictive biomarker is unlikely mainly due to the heterogenicity of the cancers and the associated resistance mechanisms. Here, we review the current state of knowledge on resistance mechanisms to combined CDK4/6 inhibitor and endocrine therapy and associated predictive molecular biomarkers, including the use of liquid biopsy.

**Abstract:**

The introduction of CDK4/6 inhibitors (CDK4/6i) in combination with endocrine therapy (ET) has revolutionized the treatment landscape for patients with estrogen receptor-positive (ER+) advanced breast cancer (ABC) and has become the new standard treatment. However, resistance to this combined therapy inevitably develops and represents a major clinical challenge in the management of ER+ ABC. Currently, elucidation of the resistance mechanisms, identification of predictive biomarkers, and development of novel effective combined targeted treatments to overcome the resistance are active areas of research. Given the heterogeneity of the resistance mechanisms towards combined CDK4/6i and ET, identification of a single universal predictive biomarker of resistance is unlikely. Novel approaches are being explored, including examination of multiple genetic alterations in circulating cell-free tumor DNA in liquid biopsies from ABC patients with disease progression on combined CDK4/6i and ET treatment. Here, we review the molecular basis of the main known resistance mechanisms towards combined CDK4/6i and ET and associated potential biomarkers. As inhibiting key molecules in the pathways driving resistance may play an important role in the selection of therapeutic strategies for patients who experience disease progression on combined CDK4/6i and ET, we also review preclinical and early phase clinical data on novel combination therapies for these patients.

## 1. Introduction

Breast cancer is the most commonly diagnosed cancer in women and the second leading cause of cancer death worldwide. In 2020, breast cancer accounted for approximately 11.7% (2.3 million cases diagnosed) of all newly diagnosed cancers worldwide [1]. Approximately 80% of all breast cancer cases are estrogen receptor-positive and human epidermal growth receptor 2 negative (ER+/HER2−). Recently, the number of treatment options for ER+/HER2− advanced breast cancer (ABC) patients has increased due to the expanding knowledge on the biological complexity of breast cancer, which has resulted in better palliative treatment regimens and improved survival [2].

In 2016, combined CDK4/6 inhibitors (CDK4/6i) and endocrine therapy (ET) was approved by the FDA and EMA for treatment of ER+/HER2− ABC patients and has become standard treatment in the first-line setting combined with an aromatase inhibitor (AI) or as second-line treatment in combination with fulvestrant following progression on initial AI monotherapy. The impressive improvement of progression-free survival (PFS) and overall survival (OS), oral availability, and relatively low toxicity profile that remarkably improve the quality of life of these patients are some of the advantages of combined CDK4/6i and ET [3,4,5,6,7,8,9,10]. Currently, three CDK4/6i, palbociclib, abemaciclib, and ribociclib are routinely used in the clinic. However, ER+/HER2− ABC patients inevitably develop resistance to combined CDK4/6i and ET, questioning whether we currently have the optimal sequence of available therapies and whether novel more effective therapies can be generated. The mechanisms underlying resistance to combined CDK4/6i and ET remain poorly understood and only limited data are available on the efficacy of other novel endocrine-based combinations, including ET with mTOR or PI3K inhibitors, following progression on combined CDK4/6i and ET. Thus, predictive biomarkers for optimal patient selection and novel drug combinations for those who develop resistance to combined CDK4/6i and ET are urgently needed.

In this review, we discuss the mechanisms of resistance, potential biomarkers, and promising novel treatment combinations to improve clinical outcomes in ER+/HER2− ABC.

## 2. Mechanisms of Resistance to Combined CDK4/6i and ET in Patients with ER+/HER2− ABC and Potential Predictive Biomarkers for Clinical Use

Both primary and acquired resistance to combined CDK4/6i and ET treatment are observed in ER+/HER2− ABC patients. Mature data from PALOMA, MONALEESA, and MONARCH clinical trials indicate that approximately 20% of patients treated with combined CDK4/6i and ET experienced primary resistance, and all patients eventually progress on treatment [11].

Several studies have attempted to identify molecular biomarkers that predict the response (sensitivity or resistance) to combined CDK4/6i and ET treatment in ABC. Although these studies have identified various candidates, none has been sufficiently validated for clinical use. Indeed, several potential biomarkers have been identified using breast cancer cell models resistant to the single-agent CDK4/6i, and not to the clinically-used combination of CDK4/6i and ET, and thus do not likely reflect the possible multi-step process of resistance wherein cells undergo early adaptation before long term resistance [12]. Nevertheless, several cell cycle-specific components and regulators of the mitogenic signaling pathways have been suggested to be involved in the mechanism of resistance to combined CDK4/6i and ET treatment, which could be utilized as resistance biomarkers. Herein, we summarize the main biological drivers of resistance to combined CDK4/6i and ET. Several of these drivers may also be used as potential new therapeutic targets and be part of the clinical treatment strategy adapted upon progression on combined CDK4/6i and ET (Figure 1).

### 2.1. Cell Cycle-Specific Components

#### 2.1.1. Loss of Rb Function

The tumor suppressor Rb protein is the direct target of cyclin-CDKs that control the G1/S phase transition of the cell cycle, mainly through binding to the E2F family of transcription factors and suppressing their activity. Thus, loss (depletion) or inactivation (mutation or hyperphosphorylation) of Rb protein may facilitate unregulated cell cycle progression at the G1/S phase transition of the cell cycle [13,14,15]. In preclinical studies, inactivation by Rb hyperphosphorylation has been detected in ET-resistant breast cancer cell models and tumors of patients receiving adjuvant ET [15]. A molecular signature of Rb loss of function (Rbsig) consisting of 87 E2F regulated genes has been developed using palbociclib-resistant breast cancer cell models and shown to be prognostic in patients with ER+ early breast cancer [16]. An association between Rb loss of function and resistance to combined CDK4/6i and ET was also observed in 3 patients with ER+/HER2− ABC who received CDK4/6i (palbociclib or ribociclib) and showed somatic mutations in Rb in circulating tumor DNA (ctDNA) upon disease progression [17]. However, the predictive value of Rb remains uncertain, as tumor analyses performed on two large randomized phase 3 clinical trials (PALOMA-2 and 3, examining palbociclib in combination with AI or fulvestrant versus AI or fulvestrant alone, respectively, in patients with ER+/HER2−ABC) failed to show a statistically significant correlation between Rb (at mRNA or protein level) and resistance to CDK4/6i [18,19]. Importantly, ctDNA analysis of patients from the PALOMA-3 trial showed that *RB1* mutations emerged only in the combined palbociclib and fulvestrant arm, but with relatively low frequency (4.7%) [20]. One possible explanation for the lack of predictive value is that evaluation of *Rb* depletion or mutations does not reflect the functionality of the protein, as hyperphosphorylation is also one of the inactivation mechanisms of Rb protein. Therefore, further investigations are warranted to detail the role of Rb loss-of-function in resistance to combined CDK4/6i and ET in larger randomized trials and its use as a predictive biomarker.

#### 2.1.2. Hyperactivity of the Cyclin E-CDK2 Axis

Increasing activity of the cyclin E-CDK2 complex has been shown to be involved in endocrine resistance [21,22]. CDK2 activation has been suggested to compensate for CDK4/6 inhibition by hyperphosphorylation of Rb, providing a possible mechanism of resistance to CDK4/6i [23]. We have recently found activation of the cyclin E-CDK2 axis in both breast cancer cell models resistant to AI monotherapy and combined CDK4/6i with fulvestrant [24]. Another study suggested modulation of MYC via activated cyclin E-CDK2 as a resistance mechanism in CDK4/6i-resistant breast cancer cell models [25]. Moreover, early adaptation of ER+ breast cancer cell models to CDK4/6i has been associated with increased activity of CDK2 by noncanonical binding with cyclin D1 mediating G1/S transition cell cycle entry [12]. Interestingly, gene expression analysis of metastatic tissue from patients enrolled in the PALOMA-3 trial showed that high CCNE1 mRNA expression was associated with early progression (resistance) to combined palbociclib and fulvestrant [18]. However, the PALOMA-2 and MONALEESA-2 trials evaluating CDK4/6i and letrozole versus letrozole alone in patients with advanced disease and no prior ET failed to demonstrate such a correlation [5,19], suggesting that the tumor level of CCNE1 mRNA might be a potential biomarker in ER+/HER2− ABC patients previously treated with ET. Although the analysis of liquid biopsy components such as tumor exosomes and circulating cell-free RNA are still at an early stage [26], future evaluation of alterations of the cyclin E1-CDK2 axis in liquid biopsies of ER+/HER2− ABC patients is promising.

#### 2.1.3. Upregulation of CDK6

Although mutations in the kinase domain of CDK4 and CDK6 have not been reported as mechanisms of resistance to CDK4/6i, *CDK6* amplification was described in CDK4/6i-resistant breast cancer cell models [27]. In contrast, *CDK4* amplification has not been found in resistance models and induced overexpression of CDK4 did not promote resistance to CDK4/6i [27]. Additionally, upregulation of CDK6 has been found to be associated with resistance to ET and also observed following long-term exposure to CDK4/6i [12,28]. Interestingly, it has been demonstrated that the increased CDK6 expression observed in CDK4/6i-resistant breast cancer cell models is dependent on the suppression of the TGF-β pathway by the miR-432-5p expression [29]. The importance of CDK6 overexpression in mediating resistance to CDK4/6i is further supported by another study in which FAT1 loss-of-function was shown to induce CDK6 overexpression via suppression of the Hippo pathway in tumors of ER+/HER2− breast cancer patients after initiation of CDK4/6i treatment [30]. It is noteworthy that the presence of *FAT1* mutation was associated with shorter PFS compared to patients with tumors without *FAT1* mutations. However, gene analysis showed that only 6% of ER+/HER2− ABC harbored FAT1 loss-of-function mutations in their metastatic tumors after treatment with CDK4/6i [30], likely representing a small population of patients with resistance to CDK4/6i, and the predictive value *FAT1* mutations remain unclear.

#### 2.1.4. Activation of TK1

Thymidine Kinase 1 (TK1), a cytosolic enzyme involved in the pyrimidine salvage pathway, is a key regulator of DNA synthesis and cell proliferation. It can be evaluated by detecting the protein level (in cells or plasma) or measuring its activity (TK_a_) [31]. Baseline plasma TK_a_ in ER+/HER2− ABC patients treated with ET has been suggested as an early promising prognostic and predictive biomarker [32,33]. Since TK1 is among the E2F target genes and CDK4/6i prevents G1/S transition of the cell cycle, it has been hypothesized that TK1 could be a biomarker of CDK4/6i efficiency. Indeed, it has been recently shown that TK1 level and TK_a_ are reduced shortly (3 days) after treatment with palbociclib in palbociclib-sensitive, but not in palbociclib-resistant, ER+ breast cancer cell models [34]. Reduction of TK_a_ has been suggested as a possible early biomarker of proliferative inhibition in response to palbociclib [34]. The prognostic value of TK_a_ has been validated clinically using plasma samples from ER+/HER2− ABC patients in the TREnd trial (NCT02549430) [35]. This trial evaluated the activity and safety of palbociclib single agent or combined with ET and found that patients with tumors with low baseline TK_a_ significantly correlated with longer PFS compared to those with high levels. Similarly, patients with increasing TK_a_ levels during treatment had a shorter time to disease progression compared to those with TK_a_ levels that remained stable or decreased in response to treatment [35]. This study suggests plasma TK_a_ levels as a dynamic biomarker that correlates with response to palbociclib in palbociclib-treated patients [35]. Additionally, overexpression of plasma TK1 (mRNA measured in plasma-derived exosomes) was also found to be associated with resistance to CDK4/6i in palbociclib- and ET-treated ER+/HER2− ABC patients [36]. Currently, the prognostic value of plasma TK_a_ assessed at the baseline and upon treatment with CDK4/6i and ET in ER+/HER2− ABC is being further evaluated in ongoing clinical trials, including BioItaLEE (NCT03439046) and PYTHIA (NCT02536742).

### 2.2. Adaptation of Mitogenic Signaling Pathways

#### 2.2.1. The PI3K/AKT/mTOR Signaling

The crosstalk between the cyclin D-CDK4/6 and PI3K/AKT/mTOR pathways has been extensively described [37]. Indeed, upregulation of PI3K/AKT/mTOR pathway has been reported in response to chronic exposure to CDK4/6i and resulted in non-canonical activation of CDK2 by binding to cyclin D and subsequently driving cell cycle progression [12]. Importantly, alterations of the PI3K pathway, particularly *PIK3CA* mutations, are found in 40% of ER+/HER2− ABC patients [38]. Recently, analysis of *PIK3CA* mutations in ctDNA of 30 ER+/HER2− ABC patients at baseline prior to combined CDK4/6i and ET treatment showed significantly worse PFS in patients with *PIK3CA* mutated ctDNA [39]. However, prospective analysis of *PIK3CA* mutations in plasma ctDNA at baseline from patients enrolled in PALOMA-3 [20] and MONALEESA-3 [40] trials showed no significant association between *PIK3CA* status and response to CDK4/6i, although lower PFS was reported in *PIK3CA*-mutated patients. One explanation is that *PIK3CA* mutations are trunk mutations occurring early in tumorigenesis and thus are present in all malignant lesions. Interestingly, a relative change in plasma *PIK3CA* ctDNA levels 15 days after treatment (15 days/baseline) strongly predicts PFS on combined palbociclib and fulvestrant in PALOMA-3 trial [41]. Further analysis of the PALOMA-3 trial assessing relative changes in ctDNA levels at further time points (30day/baseline and at progression) showed a similar correlation [42]. These results demonstrated that early ctDNA dynamics of *PIK3CA* mutation is a prognostic factor of PFS that can be used to monitor the efficacy of palbociclib and fulvestrant in patients with ER+/HER2− ABC.

Preclinically, activation of 3-phosphoinositide-dependent protein kinase 1 (PDK1) can directly activate AKT [43]. Indeed, a kinome-wide siRNA screen analysis showed that overexpression of PDK1 resulted in the activation of AKT and CDK2 in ribociclib-resistant breast cancer cell models [44]. Increased levels of activated AKT were also reported in combined palbociclib and fulvestrant-resistant breast cancer cell models and were significantly associated with low PFS in ER+/HER2− ABC patients treated with combined CDK4/6I and ET [45]. Another study has shown that loss of PTEN mediates overexpression of activated AKT, CDK4, and CDK2 in CDK4/6i-resistant breast cancer cell models [46]. The predictive value of these biomarkers requires further validation in large cohorts and randomized trials.

#### 2.2.2. ESR1 Mutations

Acquisition of *ESR1* mutations is also frequently observed (25–40%) in patients with ER+/HER2− ABC who progressed on prior ET, particularly with AI [47,48]. The majority of *ESR1* mutations are within the ligand-binding domain (LBD) of ER (D538G, Y537C, Y537N and Y537S) [47]. It has been suggested that *ESR1* mutations may usually be subclonal in the resistant breast cancer cells arising in response to ET [20,49], yet the effect of this subclonality in response to therapy is unknown. The prognostic value of *ESR1* mutations (in the tumor and cfDNA) was reported to be significant in AI-treated ER+/HER2− ABC patients, and this significance could be powered in patients receiving more lines of ET [50,51]. However, detection of plasma *ESR1*-mutated ctDNA dynamic (at baseline and the end of treatment) from patients enrolled in the PALOMA-3 trial offered a limited prediction of resistance to CDK4/6i [20]. This analysis showed that ER+/HER2− ABC patients treated with fulvestrant single agent or in combination with palbociclib frequently acquired polyclonal *ESR1* mutations, particularly Y537S, during treatment [20]. Similar results were also obtained by another analysis of patient samples from the PALOMA-3 trial, as the early measurement of plasma *ESR1*-mutated ctDNA dynamic (at baseline and 15 days of treatment) did not offer a prediction of clinical outcome [41]. These results could reflect the substantial subclonal complexity of breast cancer upon progression on ET and suggest that early and late progression have distinct mechanisms of resistance. Further development and standardization of mutation tracking in liquid biopsies are ongoing.

#### 2.2.3. Alterations of the FGFR Signaling

In preclinical studies, overexpression of *FGFR1* has been found to be correlated with resistance to combined CDK4/6i and ET [52]. Indeed, a kinome-wide siRNA screen analysis showed that resistance to single-agent ET and in combination with CDK4/6i are partially dependent on the FGFR signaling [53]. Moreover, activating *FGFR* mutations have been reported in ER+/HER2− ABC patients and suggested to be associated with resistance to the combined CDK4/6i and ET [54]. In contrast to the conflicting data on the role of alterations in other growth pathways (i.e., *PIK3CA* and *ESR1*), the association between FGFR alterations and CDK4/6i response is more clear based on the data from large randomized trials. Analysis of plasma ctDNA of ER+/HER2− ABC patients enrolled in the MONALEESA-2 trial found *FGFR1* amplification or activating mutations in 41% (14/34) in patients who had progressed on CDK4/6i [52]. Furthermore, patients harboring *FGFR1* amplification exhibited significantly shorter PFS compared with patients with *FGFR1*-wild type [52]. Recently, plasma *FGFR1* amplification in ctDNA at baseline was also found to be significantly associated with low PFS in ER+/HER2− ABC patients from the PALOMA-3 trial who progressed on CDK4/6i [55].

## 3. Tissue Source for Biomarker Analysis

The conflicting data across the different studies regarding biomarkers might be explained by several factors, including the different drug pharmacometrics, type of specimen (metastasis versus primary tumor) analyzed, and the different analytical platforms. In addition, the resistance mechanism to the combined CDK4/6i and ET are heterogeneous in nature and biomarkers have been primarily examined in the biopsy of the primary tumor at diagnosis, which makes them mainly prognostic. This is one of the main challenges for predictive biomarker discovery in the advanced setting, as biopsies of metastatic tissue collected on treatment and after disease progression are frequently unavailable and biomarkers may evolve during tumor treatment and progression. Indeed, *CCNE1* has been demonstrated in the PALOMA-3 study to be relatively predictive when assessed in pretreatment metastatic biopsies, but such alteration was only marginal in biopsies from the primary tumor obtained at diagnosis [18]. Thus, for ABC patients it is optimal to assess a selected biomarker in matched pre-treatment (right before treatment) and post-progression metastatic biopsies. Additionally, differences may also be observed when assessing a suggested biomarker in different types of specimens, such as tissue biopsies and using liquid biopsies. Although tissue biopsy is currently the method of choice to obtain tumor molecular information, it is invasive and might be affected by the tumor heterogeneity [26,56]. Thus, tissue biopsies are unable to represent the complete molecular picture of a given tumor, particularly if multiple tumor lesions are present, which is often the case in ABC. On the other hand, liquid biopsy permits the study of disease features in a more comprehensive manner by measuring the tumor components in biofluid samples [26,56]. Particularly, the analysis of mutations in the ctDNA dynamic holds huge potential in predicting treatment outcomes in ABC [57,58]. Furthermore, a broader analysis of multiple gene alterations, such as RBsig, might be a better predictor of resistance to the combined CDK4/6i and ET than a single biomarker. In line with this, we have shown that the combined protein levels of CDK6, phospho-CDK2, and cyclin E1 significantly correlated with shorter PFS on AI treatment and combined CDK4/6i and ET treatment in ER+ ABC patients [24]. Another study reported that a high CCNE1/RB1 expression ratio is associated with palbociclib resistance in a panel of ER+ breast cancer models, and the clinical utility of this expression ratio was validated in the neoadjuvant setting by retrospective analysis of the NeoPalAna trial [59]. Generally, it seems evaluation of ctDNA dynamics using multi-gene panels more efficiently predict treatment outcome [57,58].

## 4. Role of CDK4/6i after Disease Progression

Whether CDK4/6i treatment should be continued beyond the development of disease progression remains to be defined. Recent preclinical studies reported cross-resistance between the three approved CDK4/6i in palbociclib-resistant breast cancer cell models [60,61]. However, some studies suggest that a substantial number of patients continue to derive clinical benefit with abemaciclib after prior palbociclib or ribociclib treatment [62,63,64]. Two clinical studies are ongoing to determine whether ER+/HER2− ABC patients who had progressed while receiving combined palbociclib and AI would benefit from combined palbociclib and fulvestrant (NCT02738866), combined ribociclib, and fulvestrant or fulvestrant alone (NCT02632045).

## 5. Novel Combinations Strategies for ER+/HER2− ABC Patients following Progression on CDK4/6i and ET

The development of novel treatment combinations for ER+/HER2− ABC has recently become an area of active research. Based on the widely known crosstalk between the cyclin D-CDK4/6-Rb pathway and other signaling pathways and the expanding knowledge on the mechanisms of resistance to CDK4/6i, several novel combinations have been proposed (Figure 2). These include triple combinations with new targeted agents in combination with CDK4/6i and ET, which may either be administrated following progression on combined CDK4/6i and ET or as a replacement of combined CDK4/6i and ET treatment. In the latter case, clinical implementation of these triple combinations for ER+/HER2− ABC patients will require significant improvement in the PFS compared to the combined CDK4/6i and ET double treatment due to potential overlapping toxicities and financial implications.

### 5.1. Combination Treatments Containing Agents Targeting the PI3K/AKT/mTOR Pathway

#### 5.1.1. α-Specific PI3K Inhibitor Alpelisib

Alpelisib was the first α-specific PI3K inhibitor to be approved by the FDA (May 2019) and EMA (July 2020) in combination with fulvestrant as a second-line treatment for *PIK3CA*-mutated ER+/HER2− ABC after progression on AI. The FDA also recommends that patients who progressed on AI and have negative ctDNA *PIK3CA*-mutation tests should undergo tumor biopsy for *PIK3CA*-mutation assessment [65]. The approval of alpelisib in this setting was based on the results from the phase III SOLAR-1 trial that showed a significant improvement of the PFS with alpelisib plus fulvestrant compared with fulvestrant alone (11 vs. 5.7 months, *p* < 0.001) [66]. However, data from this trial also showed considerable toxicity in the alpelisib arm, including hyperglycemia (64%), rash and diarrhea (58%), and nausea (45%), with 5% of the patients discontinuing treatment [67]. Importantly, the efficacy of PI3Ki upon progression on CDK4/6i has been observed in preclinical and clinical studies. Indeed, several preclinical studies have shown synergistic inhibition of Rb and mTORC1 by combining PI3Ki and CDK4/6i in breast cancer cell models resistant to CDK4/6i single agent [12,44]. Although the SOLAR-1 study included a small subset of patients (5.9%) previously treated with CDK4/6i, the median PFS was 5.5 months in the alpelisib plus fulvestrant arm compared to 1.8 months in the fulvestrant arm. However, the phase II BYLieve study (NCT030556755) is the first trial assessing the efficacy of alpelisib and ET in patients with *PIK3CA*-mutated ER+/HER2− ABC who progressed on or after therapy with a CDK4/6i. This trial recently met its primary endpoint, with 50.4% of the patients showing no disease progression at 6 months and a median PFS of 7.3 months [68]. Although this combination may be efficacious in patients with *PIK3CA* mutations that were previously treated with CDK4/6i, a firm conclusion would be premature. Interestingly, a recent preclinical study performed by our group reported a limited efficacy of this combination upon progression on CDK4/6i based therapy [69]. However, recent preclinical studies suggested that triple combination treatment with ET, CDK4/6i and PI3Ki may overcome resistance to single-agent CDK4/6i [70]. Although the triple combination treatment approach may increase the efficacy of the targeted agents, it is associated with increased toxicity. Therefore, several trials are ongoing investigating the safety and efficacy of the triple combination with ET, CDK4/6i, and PI3Ki (Table 1). Other studies have suggested sequential rather than simultaneous exposure to these agents, offering PI3Ki to patients who harbored *PIK3CA* mutation following treatment with a CDK4/6 inhibitor [70].

#### 5.1.2. mTOR Inhibitor Everolimus

Everolimus was the first mTOR inhibitor to be approved by the FDA in combination with steroidal AI for the treatment of ER+/HER2− ABC patients who progressed after exposure to non-steroidal AI as first-line therapy [71]. This approval was based on the results from phase III BOLERO-2 trial showing an improved PFS in ER+/HER2− ABC patients treated with everolimus plus exemestane compared with exemestane alone (11.0 vs. 4.1 months, *p* < 0.0001). However, everolimus treatment is associated with severe side effects, including stomatitis (59%), rash (39%), hyperglycemia (37%), diarrhea (34%), and fatigue (4%), which has resulted in the limited clinical use of this drug combination. Interestingly, a small clinical study found limited efficacy of CDK4/6i combinations after progression with everolimus therapy [72]. However, a preclinical study showed that treatment of CDK4/6i and ET-resistant ER+/HER2− breast cancer models with everolimus restored sensitivity to CDK4/6i [73], suggesting that combining these therapies may be an option after progression on CDK4/6i71. Several ongoing trials are addressing the efficacy of the triple combination with ET, CDK4/6i, and mTOR inhibitors following progression on CDK4/6i-based therapy (Table 1).

### 5.2. Combination of CDK4/6i and Immunotherapy

Several preclinical studies have recently reported the ability of CDK4/6i to induce anti-tumor immune responses [74,75] by enhancing tumor antigen presentation major histocompatibility complex (MHC), increasing production of type III interferons, and by reducing the proliferation of immune-suppressive regulatory T-cells (Treg-cells) [74]. These effects of CDK4/6i are associated with inhibition of the E2F target, DNA methyltransferase 1 [74]. Additionally, CDK4/6i can enhance the activity of effector T-cells through activation of NFAT transcription factors and their targeted genes, particularly IL-2 [75]. Furthermore, CDK4/6i have been shown to augment the response to anti-PD-1 therapy [75]. Based on these data, several ongoing clinical studies are investigating the CDK4/6i in combination with anti-PD-1/PD-L1 inhibitors in addition to ET (AI or fulvestrant). Most of these clinical studies are evaluating the efficacy of this triple combination in patients without prior exposure to CDK4/6i (NCT02778685 and NCT03294649), whereas the phase II PACE clinical study (NCT03147287) is assessing the efficacy of combined avelumab (PD-L1 inhibitor), fulvestrant and palbociclib on metastatic patients who experienced disease progression on CDK4/6i-based therapy.

### 5.3. Novel Endocrine Therapies

As mentioned earlier, breast cancer cells acquire *ESR1* mutation following exposure to ET, particularly an AI, and also in combination with CDK4/6i [20,47,48,76]. Although mutant *ESR1* breast cancer cells show resistance to AI, they retain relative sensitivity to fulvestrant, as reported in the phase III soFEA trial (NCT00253422). This study demonstrated improvement in the PFS with fulvestrant over AI in ER+/HER2− ABC patients with mutated *ESR1* ctDNA [77]. However, different mutations correlated with different levels of responses to fulvestrant (e.g., ER Y537S mutation was more resistant than D538G), as shown in vitro [50]. Moreover, *ESR1* Y537S mutant ctDNA could be acquired at disease progression with both fulvestrant single agent and combined with CDK4/6i, as shown in the PALOMA-3 trial [20].

Various strategies that target mutant *ESR1* breast cancer, including novel estrogen receptor down regulator (SERD) or estrogen receptor modulator (SERM) orally bioavailable, are in clinical development [78]. Among these agents, GLL398 was found to be a highly potent SERD for Y537S-mutated ER expressing tamoxifen-resistant breast cancer cell models [79]. Other agents entered early clinical trials, such as LSZ102 (NCT02734615) combined with CDK4/6i in ER+/HER2− ABC patients who progressed on ET, and the oral SERD elacestrant (RAD1901, NCT03778931) [80] in ER+/HER2− ABC patients who experienced disease progression on combined CDK4/6i and ET (AI or fulvestrant).

Interestingly, a novel compound, ErSO, represents an alternative strategy to the inhibition-based approaches of ET [81]. This molecule acts through leveraging ER expression, which is typically maintained in ET-resistant breast cancer cells, via over-activating the tumor-protective ER-activated anticipatory unfolded protein response (a-UPR) in breast cancer cells to initiate cell death. Rapid and selective necrosis in ER+ breast cancer cell models and nearly a complete regression of tumors in vivo (cell models, patient-derived orthotopic and metastatic xenografts) have been observed. Clinical trials will be initiated in the near future [81].

### 5.4. Combination of CDK4/6i with Other Targeted Agents

#### 5.4.1. CDK2 Inhibitors

Preclinical studies have demonstrated the involvement of activated cyclin E-CDK2 in CDK4/6i-resistance ER+/HER2− ABC [21,22,23]. Recently, we found that co-targeting CDK2, CDK6, and ER signaling inhibits the growth of ER+/HER2− breast cancer cells resistant to combined CDK4/6i and ET [24], supporting the use of this triple combination as a novel therapeutic strategy. However, there are no specific CDK2 inhibitors approved for clinical use. Nevertheless, a novel potent and selective CDK2/4/6 inhibitor, PF-06873600, was recently developed and demonstrated activity in Rb-deficient palbociclib-resistant breast cancer cell models exhibiting high cyclin E expression [82]. An ongoing phase I trial is assessing the efficacy of PF-06873600 as a single agent and in combination with ET as first-line treatment in ER+/HER2− ABC patients (NCT03519178). Another CDK2 inhibitor, BLU-222 from Blueprint Medicines is currently undergoing development at MD Anderson Cancer Center.

#### 5.4.2. FGFR Inhibitors

FGFR has emerged as a new therapeutic target following the preclinical data reporting the involvement of *FGFR1* amplification/mutation in the mechanisms of resistance to both ET and CDK4/6i [54]. Recently, the maximum tolerated dosage (MTD) of the FGFR inhibitor BGJ398 in combination with α-specific PI3K inhibitor alpelisib in *PIK3CA* mutant-solid tumors was determined [83]. An ongoing phase I trial is addressing the efficacy of erdafitinib, an FGFR inhibitor approved by the FDA for urothelial cancer, combined with CDK4/6i and fulvestrant in ER+/HER2− ABC patients who harbor *FGFR* amplification and progressed on first-line therapy, including patients previously exposed to CDK4/6i (NCT03238196).

#### 5.4.3. CDK7 Inhibitors

CDK7, a transcriptional CDK exhibiting CDK activating kinase activity on CDK2 and CDK9, has recently been found to be involved in the resistance mechanisms to CDK4/6i in CDK4/6i-resistant ER+/HER2− breast cancer cell models with loss of Rb and exhibiting cyclin E1 overexpression [84]. Furthermore, synergistic antitumor activity was observed in these models when combining the CDK7 inhibitor SY-1365 with fulvestrant [84]. Ongoing phase I trials are investigating the efficacy of the CDK7 inhibitors SY-1365 and CT-7001 combined with fulvestrant in patients with solid tumors, including ER+/HER2− breast cancer patients who progressed on CDK4/6i-based therapy (NCT03134638, and NCT03363893).

#### 5.4.4. BCL2 Inhibitor

BCL2 is an estrogen-responsive gene and anti-apoptotic molecule that is overexpressed in about 80% of BC [85]. The selective BCL2 inhibitor ABT-199 (venetoclax), approved by the FDA for acute myeloid leukemia and chronic lymphocytic leukemia [86], demonstrated notable activity in a phase I clinical trial when combined with tamoxifen in ER+/HER2− ABC patients with BCL2 overexpression [87]. Furthermore, the ongoing phase I clinical study PALVEN (NCT03900884) is evaluating the safety and efficacy of the triple combination with CDK4/6i, AI letrozole, and ABT-199 as first-line treatment in ER+/HER2− ABC patients with BCL2 overexpression. Additionally, the ongoing phase II clinical trial VERONICA (NCT03584009), is assessing the efficacy of ABT-199 in combination with fulvestrant in patients who progressed on CDK4/6i-based therapy.

#### 5.4.5. Mitotic Kinases Inhibitors

Aurora kinases A, B and C are mitotic kinases regulating chromosomal segregation that have been shown to be upregulated in breast cancer [88]. Notably, loss of Rb has been associated with centromere dysfunction and chromosomal instability [13]. Recently, a preclinical study reported a lethal interaction between an aurora A inhibitor and loss-of-function mutations in Rb in a panel of cancer cells, including CDK4/6i-resistant breast cancer cell models [89]. The aurora A selective inhibitor LY3295668 (Erbumine) is currently being tested in a phase I clinical trial (NCT03955939) as a single agent and combined with ET in ER+/HER2− ABC patients previously treated with CDK4/6i-based therapy.

## 6. Conclusions

Combined CDK4/6i and ET is a key factor in the therapeutic strategies for the management of ER+/HER2− ABC patients and has become the treatment of choice in the first- and later-line settings, but resistance eventually develops. Several important questions remain to be answered, including the diverse mechanisms involved in treatment failure, identifying biomarkers that can predict the outcome of this treatment, and determination of which novel treatment combinations can be included after progression with this combination. A possible correlation between genomic alterations and disease progression could be revealed by taking the advantage of liquid biopsy analysis. Ongoing and future clinical studies with appropriate patient selection based on biomarkers detected by state-of-the-art methods may provide the evidence needed to optimize individualized therapeutic strategies and thus further improve the outcome of ER+/HER2− ABC patients.

## Figures and Tables

**Figure 1 cancers-13-05397-f001:**
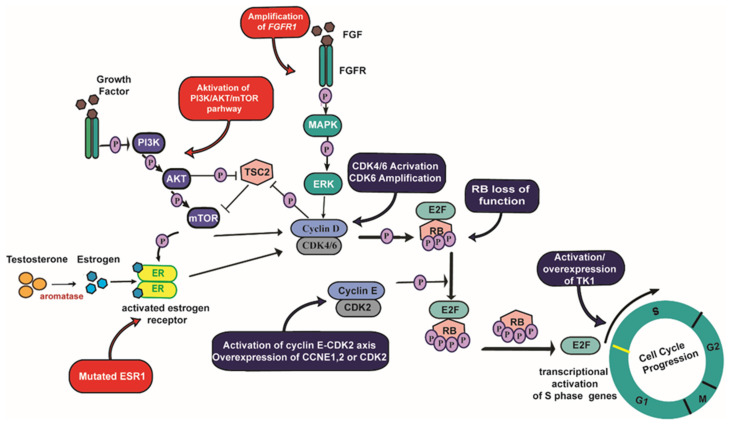
Key mechanisms implicated in the development of resistance to combined CDK4/6i and ET. Cell cycle-specific components are indicated in blue and activation of oncogenic pathways are indicated in red.

**Figure 2 cancers-13-05397-f002:**
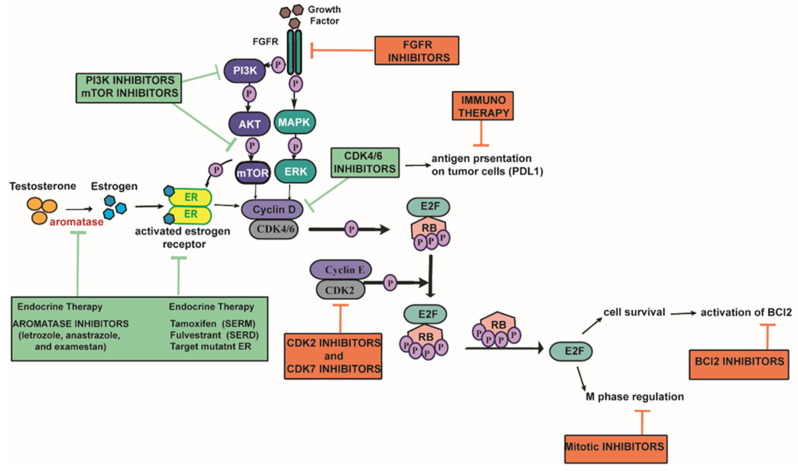
Signaling pathways associated with resistance to combined CDK4/6i and ET and promising combined treatments to overcome this resistance in ER+/HER2− ABC. Treatments in green boxes are already approved by the FDA and EMA for ER+/HER2− ABC. Pink boxes include promising treatments that are under preclinical and clinical studies.

**Table 1 cancers-13-05397-t001:** Ongoing clinical trials investigating the triple combination with approved CDK4/6, PI3K/mTOR inhibitors, and ET for ER+/HER2− ABC patients.

Target	Inhibitor	NCT Number	Phase	Estimated or Actual Participants	Setting	Prior Treatment				Backbone Treatment (ET with CDK4/6i)
						CT for ABC	ET	CDK4/6i	PI3K/mTORi	
PI3K P110-α	Alpelisilib	01872260	I, II	253	1. line and 2. line	˂1	any	Per (PhI) Not Per (PhII)	PER (PhI) Not Per (PhII)	NSAI and ribociclib
PI3K	Alpelisilib or Buparlisib *	02088684	I, II	70	1. line and 2. line	˂2 (PhI) ˂1 (PhII)	any	Per (PhI) Not Per (PhII)	Per (PhI) Not Per (PhII)	Fulvestrant and ribociclib
mTOR	Everolimus	01857193	I	132	1. line and 2. line	≤1	any	Per	Not Per	NSAI and ribociclib
mTOR	Everolimus	TRINITI 02732119	I	51	2. line	≤1	any	Per	Not Per	NSAI and ribociclib
mTOR	Everolimus	02871791	I, II	32	2. line	≤1	any	Per	Not Per	NSAI and palbociclib

* pan-PI3K inhibitor in phase III trials, NSAI: non-steroidal aromatase inhibitors; CT: chemotherapy; Per: permitted; Not Per: not permitted.

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
