# Peer review of "Resistance Mechanisms to Combined CDK4/6 Inhibitors and Endocrine Therapy in ER+/HER2− Advanced Breast Cancer: Biomarkers and Potential Novel Treatment Strategies"

_cancers, 2021, doi:10.3390/cancers13215397_

Round 1

Reviewer 1 Report

This review aims to clarify resistance mechanisms as well as the potential biomarkers that impact treatment outcomes for patient with ER+/HER2 advanced breast cancer. The review is clearly organized and the need for it is timely and well-justified in the summary, abstract and introduction.  The Figures elucidate complex interacting mechanisms to promote reader understanding. I believe the overall work makes a needed contribution to understanding the phenomenon of interest and therefore provide my brief comments in support of its publication.

Introduction: The motivation of the review is clear, the significance of specifying advanced breast cancer based on prevalence and especially the presence of ER+/HER2 cases.  In particular, it is relevantly emphasized and appreciated that the background highlights an urgent need to promote progression free survival by understanding the mechanisms and biomarkers to inform future treatment strategies.

Section 2: This section is clearly introduced and comprehensively highlights several relevant key resistance mechanisms with proper citation to the literature that supports their importance

Section 3: I appreciated the emphasis on contradictory findings and the effort the authors took to explain the open questions and their own research that supports the appropriate assay for more efficient prediction of outcomes

Section 4: Brief but appropriate considering the literature is still emerging.

Section 5: A very important contribution overall. Comprehensive references to the broader literature and the authors own work allows in depth understanding of the complexities for combining therapies while also promoting readers confidence that this is a manuscript from a group of reference.  

Overall the need for and elaboration of this literature review is well elaborated on the factors of interest regarding sensitivity or resistance and potential therapies that can be of great interest to support future treatment strategies that improve progression free survival in this population.

Author Response

We appreciate that the reviewer finds our review to be “a very important contribution overall”, and that it “clearly organized, timely and well-justified in the summary, abstract and introduction.” We are also happy that the reviewer finds that “the figures included promote reader understanding” and “elucidate the complex interacting mechanisms”. We have performed the requested spellcheck and corrected a few spelling mistakes.

Reviewer 2 Report

Summary:

This study describes the current learnings on resistance mechanisms to combine CDK4/6 inhibitor and endocrine therapy and the relationship with predictive molecular biomarkers. Moreover, also the authors make a comprehensive analysis on development of novel combination strategies for patients progressing to CDK4/6 and anti-hormone therapies.

Overall, the work is very well presented, easier to follow, and well connected with the clinical trials ongoing. I fully support the publication.

Minor comments:

The authors mentioned combined CDK4/6 inhibitor and endocrine therapy improve both progression-free and overall survival (simple summary). However, maybe the authors would like to consider to point out while although all of the shown PFS, just two of them, abemaciclib and ribociclib, so far, shown overall survival.

Also, in the section about CDK2 inhibitors, maybe it is worth to highlight other additional CDK2 inhibitor like BLU-222 from Blueprint Medicines, which recently signed an agreement with MD Anderson to accelerate the development of this molecule.

Author Response

We appreciate that the reviewer finds our “work is very well presented, easier to follow, and well connected with the clinical trials ongoing. We are also happy that the reviewer “fully support the publication” of our review.

Minor comments:

We agree the reviewer that overall survival data for combined palbociclib and AI has not yet been reported, however survival data for combined palbociclib and fulvestrant has been reported. We only mention overall survival for CDK4/6i in general and find it beyond the scope of this review to go into details about overall survival for each individual CDK4/6i’s in combination with AI and fulvestrant, respectively.

As suggested by the reviewer we have added information about BLU-222 in the section regarding CDK2 inhibitors.

Reviewer 3 Report

The authors are clearly experts in molecular mechanisms, especially those related to cell cycle progression/mitosis, and the mutations (or other genetic alterations9 that can arise in cancer cells, providing resistance to anti-cell cycle drugs... like the CDK4/6 inhibitors that are outlined here. What is interesting for the reader are the logical explanations for why certain proteins may or may not be relevant biomarkers, such as the explanation that mutated Rb genes are not the only alteration that can occur in cancers, and that also the protein can be targeted by hyperphosphorylation. This kind of logic is very instructive for students in the field and is a good example of how biomarker studies should be designed, or what should be the rationale behind their design. 

This information is provided in the review, in a usually very clearly comprehensible and well-researched fashion; also the narrative is clear and straightforward, there appear to be no gaps or deviations or losing focus of the manuscript in irrelevant, lengthy sidelines. 

The different paragraphs are all well researched and supported by very significant references, most of them from recent years. Furthermore, the authors are well informed of the ongoing clinical trials, thus providing them not only with the necessary background knowledge, but also with the rationale for WHAT EXACTLY biomarkers would be needed - and where to look for clinical applications. Again, this provides all the necessary logic for those who are new to this field (and may want to read a review like this one as an introduction). 

The narrative is built up in a logical fashion, too: first, we are introduced to the closed, usual suspects for developing resistance - and that these are logical candidates for biomarkers that correlate with drug efficacy. Then, the review is gradually drifting towards other, not directly related "rescue pathways", such as TK1 and PI3K/AKT/mTOR signaling. These are highly relevant targets for drugs themselves and are not far removed from CDK4/6 functions. 

Gradually, more remote - but always breast-cancer relevant mechanisms are introduced, like estrogen signaling (paragraph 2.2.2 ) and FGFR functions (2.2.3). 

The cartoons does not provide much novel stuff (Fig. 1 and 2). However, they do actually mention the kind of alterations, such as mutations, amplifications and deletions that are involved in resistance mechanisms, and how they may become relevant and informative biomarkers. That's the main novelty here. This could probably be even more expanded. 

Eventually, the article moves away from the biomarker question and drifts towards the use of clinical materials for this purpose. 

The last section deals with NOVEL treatments and drugs, and follows the same logical order as the first part of the manuscript. The new treatment options are introduced in the same order as the biomarkers. This makes the reading easier and less confusing. 

So about half of the article is actually not focused on biomarkers, but on treatments, including novel ones, and variations of existing ones. But this is all perfectly well covered in the title of the manuscript, and the abstract. 

I think the article clearly targets clinicians interested in innovative therapies and their molecular basis; and translational cancer researchers that are interested in clinical aspects of trials etc.  There is a bit for everyone in this article. 

Author Response

We appreciate that the reviewer finds that the review provides “logical explanations for why certain proteins may or may not be relevant biomarkers” and that “This kind of logic is very instructive for students in the field and is a good example of how biomarker studies should be designed, or what should be the rationale behind their design. “We are also happy that the reviewer finds that our review is written “in a usually very clearly comprehensible and well-researched fashion” and that “the article clearly targets clinicians interested in innovative therapies and their molecular basis; and translational cancer researchers that are interested in clinical aspects of trials etc.” We have performed the requested spellcheck and corrected a few spelling mistakes.